# Humidity Driven Transition from Insulator to Ionic Conductor in Portland Cement

**DOI:** 10.3390/ma12223701

**Published:** 2019-11-09

**Authors:** Masahiro Nagao, Kazuyo Kobayashi, Tetsuya Hori, Yaorong Li, Takashi Hibino

**Affiliations:** Graduate School of Environmental Studies, Nagoya University, Nagoya 464-8601, Japan; nagao@urban.env.nagoya-u.ac.jp (M.N.); kkoba@urban.env.nagoya-u.ac.jp (K.K.); tetsuya.hori.j6b@soken-labs.co.jp (T.H.); tmhibino@watch.ocn.ne.jp (Y.L.)

**Keywords:** Portland cement, ionic conduction, hydroxide ion, CO_2_ exposure

## Abstract

This work aims to assess ionic conduction in anhydrous cement particles and hydrated cement pastes with aging periods of 5–25 days. When a cement sample was humidified (relative humidity = 100%) over the range of 50–100 °C, it exhibited bulk conductivities of 10^−3^–10^−2^ S cm^−1^, regardless of the hydration level, whereas the interfacial conductivities varied in the range of 10^−7^–10^−3^ S cm^−1^, depending on the structural defects or conduction pathways of the sample. Both the bulk and interfacial conductivities were increased to 0.01 S cm^−1^ or higher at 100 °C, although the sample required previous moistening with water mist. The major charge carrier in the sample was determined to be hydroxide ions, and the total ion transport number was approximately 1. Exposing the sample to a mixture of carbon dioxide and water vapor caused a decrease in the bulk and interfacial conductivities; however, the bulk conductivity was returned to the initial value by treatment with an acid.

## 1. Introduction

Portland cement is the most commonly used binder in the construction industry. The major components of Portland cement are tricalcium silicate ((CaO)_3_·SiO_2_; C_3_S), dicalcium silicate ((CaO)_2_·SiO_2_; C_2_S), tricalcium aluminate ((CaO)_3_·Al_2_O_3_; C_3_A), and tetracalcium aluminoferrite ((CaO)_4_·Al_2_O_3_·Fe_2_O_3_; C_4_AF) [1,2]. C_3_S and C_2_S react with water to form the corresponding calcium silicate hydrates (C-S-H) and calcium hydroxide (Ca(OH)_2_) [3,4]. These hydration reactions proceed from the surface into the center of the cement particles. The hydration rate of C_3_S is faster than that of C_2_S due to its higher reactivity for hydration [5]. C_3_A and C_4_AF also undergo hydration, but have not been considered to produce any Ca(OH)_2_ until recently [6,7]. Similar to sodium hydroxide (NaOH) and potassium hydroxide (KOH) present in the hydrated cement paste, the produced Ca(OH)_2_ results in a pH level (11–13) in the cement paste, until these metal hydroxides are depleted by carbonation with carbon dioxide (CO_2_) that penetrates from the atmosphere [8,9].

Such alkaline characteristics have been mainly studied to elucidation of the mechanism for the formation of C-S-H [10,11,12,13]. Nevertheless, there have been few reports on ionic conduction associated with the alkalinity of Portland cement, although there was a report that describes the conduction of potassium ions in the commercial geopolymer cement known as geocement [14]. So far, electrochemical impedance spectroscopy (EIS) has been limited to assessment of the dielectric properties, mechanical strength, carbonation, and chlorination behaviors of concrete, mortar, and cement paste [15,16,17,18]. This is because the electrical resistance of hardened Portland cement pastes is too high when measured with a thick test sample under dry conditions at room temperature. In the case of solid acids (e.g., Nafion and inorganic materials), resistance measurements are typically conducted with a thin film or membrane sample in an atmosphere saturated with water vapor (relative humidity = 100%) over the temperature range from room temperature to 100 °C [19,20,21]. Operation at high humidity and temperature further enhances the proton conductivity of the solid acids [22,23]. A similar approach could be an effective technique for the measurement of cement particles and hydrated cement pastes. Data from EIS measurements, combined with X-ray crystallography and electron microscopy, would also enable postulation of the ionic conduction mechanisms in the bulk and interfacial phases of cement.

Anion exchange membranes are regarded as promising electrolytes for fuel cell and electrolysis applications in alkaline media [24,25,26]; however, their main chain and functional groups are easily degraded by an attack from hydroxide ions (OH^−^), especially at high pH and temperature [27,28,29,30]. If cement particles or paste exhibit high hydroxide ion conductivity, this material could be an alternative electrolyte to anion exchange membranes because of their high stability under conditions of high pH and temperature. In this study, we assessed the ion conducting properties of three cement samples with different degrees of hydration using the same approaches as that for the measurement of solid acids. The main contents of this work are as follows. The origin of ionic conduction associated with Ca(OH)_2_ was independently evaluated just after production and after 5 days or more. The charge carrier in the cement was determined from the element analysis at the anode and cathode sides after direct current (DC) polarization. The ion transport number of the cement was estimated by the gas concentration cell method. Finally, the effect of CO_2_ in the atmosphere on the ionic conductivity was investigated, along with the recovery of carbonated cement.

## 2. Materials and Methods 

### 2.1. Materials

The Portland cement used in this work was purchased from Taiheiyo Cement Corporation (Tokyo, Japan). Hydration of the cement was performed by the addition of deionized water to obtain a water to cement weight ratio (W/C) of 5:15, which gives a good balance between the strength and porosity [31]. The degree of hydration was adjusted by control of the mixing time from 5 to 25 days in a ball mill (P-5, Fritsch, Idar-Oberstein, Germany). The powder (0.200 g) of the anhydrous or hydrated cement was mixed with polytetrafluoroethylene (PTFE) powder (0.013 g) using a mortar and pestle, and then cold-rolled to a thickness of 300 μm and a diameter of 17 mm using a laboratory rolling mill. A hardened hydrated cement paste cube (10 mm × 10 mm × 0.5 mm) was also prepared with the same W/C for comparison. The thickness of 0.5 mm was obtained by polishing the sample surface with 100- and 200-grit SiC abrasive papers (Sankyo Rikagaku, Saitama, Japan). A Pt/C electrode (Electrochem, Woburn, MA, USA, Pt loading: 2 mg cm^−2^) was used as the electrode for all trials. High purity Ar, air, and CO_2_ were obtained from Taiyo Nippon Sanso (Japan) and high purity H_2_ was obtained from GL Science (Japan). Various mixtures of Ar and air, H_2_, or CO_2_ were produced using a mass flow controller (MGL-2, GL Science, Tokyo, Japan). Acetic acid (Wako Chemicals, Osaka, Japan) was used without further purification.

### 2.2. Characterization

The crystalline structures of the cement samples were analyzed using X-ray diffraction (XRD; Bruker D8 Advance, Bruker AXS, Karlsruhe, Germany) with Cu Kα radiation (λ = 1.5418 Å) as the X-ray source. The diffractometer was operated at 40 kV and 40 mA. Rietveld refinement of the sample was performed using the Bruker AXS’s Topas 3.0 software. Compositional mappings of the cement samples were acquired in combination with the morphological features using energy dispersive X-ray (EDX) spectroscopy and scanning electron microscopy (SEM; Jeol JSM-6610A, Tokyo, Japan) with acceleration voltages of 15 and 2 kV, respectively. Preparation of cross-sectional cement samples was performed using a beam of accelerated Ar ions rather than water to avoid damage during the cutting and polishing processes. The degree of hydration for the cement samples was calculated from the amount of chemically bound water in the paste, which was measured as the difference in weight between the samples after drying at 105 and 1000 °C, as described elsewhere [32]. The weight measurements were done using thermogravimetry (TG; Shimadze TGD-60, Kyoto, Japan). The pH of the cement sample immediately after humidification was determined using broad range pH paper (PEHANON, Macherey–Nagel, Düren, Germany). A sealed container containing sodium hydroxide particles was used to dry the cement samples with the temperature maintained at 50 °C.

### 2.3. Electrochemical Measurements

EIS measurements of the cement samples were performed using the four probe method. A cement sample was sandwiched between two Pt/C electrodes (effective area: 0.5 cm^2^) and then set in a flow of Ar saturated with water vapor (relative humidity = 100%) at temperatures between 50 and 100 °C, unless otherwise stated. Impedance spectra were collected at open-circuit voltage with an amplitude of 60 mV over the frequency range of 1–10^6^ Hz. An equivalent circuit model was used for fitting of the impedance data. DC polarization was conducted with a current density of 0.4 mA cm^−2^ in a flow of humidified air at 70 °C. The following gas concentration cells were fabricated using cement samples as the electrolyte with two Pt/C electrodes: air + Ar (0.31 atm water vapor), Pt/C| electrolyte |Pt/C, air (0.31 atm water vapor), and H_2_ (0.31 atm water vapor) Pt/C| electrolyte |Pt/C, H_2_ + Ar (0.31 atm water vapor) [33,34]. The electromotive force (EMF) of the galvanic cells was recorded at 70 °C. All measurements were conducted using a potentio-galvanostat (Solartron 1287, UK) and a frequency response analyzer (Solartron 1260, Hampshire, UK).

## 3. Results and Discussion

### 3.1. Ionic Conductivity of Portland Cements with Different Degrees of Hydration

First, the electrical resistance of the anhydrous cement sample was measured to clarify whether the cement particles become ionically conductive only by humidification. The impedance spectra recorded at 50 °C are shown in Figure 1a, in which the partial pressure of water vapor (*P*_H2O_) in the atmosphere was 0.03 atm at zero minutes and subsequently 0.12 atm. All the obtained spectra showed intercepts of the impedance lines and the real axis, circular arcs above the real axis, and inclined lines. These lines were fitted with an equivalent circuit consisting of three impedance components in series: the serial resistance, the parallel resistance, the constant phase element (CPE), and the Warburg impedance element. The serial resistance is related to short-range ion transport [35,36], which was assigned to the bulk resistance of the sample. The non-capacitive behavior is due to reversible ion transport on the surface of the cement particles. The parallel resistance is related to long-range ion transport [35,36]; however, it is not evident whether this is dominated by a grain boundary- or pore-related mechanism. This interfacial resistance component is dependent on the morphology of the cement (e.g., particle size and structural defects) [37], as will be discussed later. Figure 1b shows plots of the bulk and interfacial resistances computed from the equivalent circuit against the time after switching *P*_H2O_. The two resistances at zero minutes were obtained after previously aging at *P*_H2O_ = 0.03 atm and 50 °C (exactly speaking, the sample was already hydrated at this stage). When *P*_H2O_ was switched to 0.12 atm, the two resistances rapidly decreased with time and became almost constant at 30 min, which indicates a large contribution from ionic conduction associated with water vapor to the electrical characteristics of the cement particles.

After the impedance measurements, the cement sample was dried in a sealed container with sodium hydroxide particles. The XRD pattern of this sample was compared with that of the anhydrous cement sample, and the result is shown in Figure 1c. No significant difference in peak intensity was apparent between the two samples, except that the diffraction peaks due to gypsum disappeared after the impedance measurement (the virgin cement particles contain small amounts of CaSO_4_·2H_2_O and alkaline metal oxides (Na_2_O and K_2_O) [38]). No detectable level of crystalline Ca(OH)_2_ was generated in the cement after humidification and then drying. To understand the significant decrease in electrical resistance by the increase in *P*_H2O_ observed in Figure 1a,b, the basicity of the cement sample immediately after humidification was inspected with a pH indicator strip. The strip was colored between pink and purple by contact with a small amount of water remaining on the sample surface, which indicated a pH of 11–12, and thus, the presence of OH^−^ ions on the surface of the sample just after humidification.

The cement hydration process consists of complicated chemical reactions. First, the alkaline metal oxides are hydrated to form the corresponding metal hydroxides (NaOH and KOH). At the next stage, oxygen ions (O^2−^) of the CaO lattice in C_3_S react with protons (H^+^) in water molecules to form OH^−^ ions according to Reaction (1). The two OH^−^ ions combine with a calcium ion (Ca^2+^) of the CaO lattice to form Ca(OH)_2_ according to Reaction (2) [39].
O^2−^ (lattice) + H^+^ (aq) → OH^−^ (aq).(1)
2OH^−^ (aq) + Ca^2+^ (aq) ⇄ Ca(OH)_2_ (solid).(2)

These rapid reactions proceed for a few hours in what is called a pre-induction process [40]. It is thus assumed that the decrease in the electrical resistance of the cement sample by humidification is due to the formation of OH^−^, Na^+^, K^+^, and Ca^2+^ ions. In addition, conduction pathways are established by water channels on the surface of C_3_S particles or the pore wall of C-S-H, in which the produced ions migrate, probably via the vehicular diffusion mechanism [41,42,43].

Ionic conduction in two paste samples with different degrees of hydration (0.33 and 0.99) was investigated next. XRD patterns of these paste samples are provided in Figure 2a, including that of the anhydrous sample for comparison. Although peaks that correspond to C_3_S, C_2_S, C_3_A, and C_4_FA were observed for all three samples, their intensities gradually decreased as the hydration proceeded. The progress of hydration was also confirmed by the appearance of new peaks identified as crystalline Ca(OH)_2_ for the two paste samples, the intensity of which increased with the hydration process. The paste samples also showed a distinct peak at a low Bragg angle around 9°, which may be attributed to C-S-H [44,45].

The temperature dependence of the electrical conductivity for the three cement samples was measured over the temperature range of 50–100 °C, and the results are presented in Figure 2b. The bulk and interfacial conductivities were calculated using the following equation:*σ*_Bulk or Interf_ = *t*/(*R*_Bulk or Interf_ × *S*),(3)
where *σ*_Bulk_, *σ_Interf_*, *R*_Bulk_, and *R*_Interf_ denote the conductivities and resistances of the bulk and grain boundary or pore wall phase of the sample, respectively, *t* is the thickness of the sample, and *S* is the surface area of electrode. Despite the different degrees of hydration, the three cement samples showed similar bulk conductivities at each temperature, so that their activation energies for ionic conduction were estimated to be 0.39–0.54 eV. In contrast, the interfacial conductivity was dependent on the sample species: anhydrous cement particles (0.70 eV) = cement paste hydrated for 5 days (0.66 eV) > cement paste hydrated for 25 days (1.53 eV, hereafter denoted as highly hydrated sample). This difference was closely related to structural defects rather than the particle size of the samples, as shown in Figure 3, which indicated that the particle sizes of the three samples were comparable across the samples, while cracks were grown, especially in the highly hydrated sample. Similar low interfacial conductivities were observed for the hardened hydrated cement paste sample, in which a number of micrometer size holes were present. Structural defects, such as cracks and holes, intercept long-range ion transport, which results in low interfacial conductivity. Alternatively, the conduction pathway shifts from the grain boundaries of C_3_S and C_2_S particles to the pore walls of meso- or microporous C-S-H as a consequence of a significant hydration process.

No significant difference was observed between the bulk conductivities of the anhydrous and hydrated cement samples, which suggests that crystalline Ca(OH)_2_ does not contribute to ionic conduction in the bulk of the three samples. Evidence for this suggestion is provided as follows. Water mist was sprayed onto the highly hydrated sample to obtain a W/C of 3:17. The XRD pattern and temperature dependence of the conductivity for this sample are shown in Figure 4a,b, respectively, along with data for the same sample with different pretreatments. The intensities of the XRD peaks attributed to Ca(OH)_2_ were found to be considerably decreased for the water-added sample. This sample showed much higher bulk and interfacial conductivities with much lower activation energies of ca. 0.1 eV, compared to those of the untreated sample, over the temperature range of 50–100 °C. These results demonstrate that the dissolved Ca(OH)_2_ and the added water enhanced ionic conduction in the highly hydrated sample. In particular, the residual water was effective for improving the interfacial conductivity, which was achieved by the expansion of water channels on the particle surface or the pore walls. This may also transfer the vehicular diffusion mechanism to the Grotthuss mechanism [41,42,43]. This sample was subsequently dried in the container with sodium hydroxide, followed by XRD and conductivity measurements. As a consequence, the dissolved Ca(OH)_2_ was recrystallized, and both the bulk and interfacial conductivities were decreased. Based on these observations, crystalline Ca(OH)_2_ must be dissolved in excess water to be used as the source of charge carriers, the condition of which corresponds to the supersaturation of water vapor (relative humidity > 100%). While it is difficult for gas-phase systems to sustain such a condition over the long term, this problem can be avoided in aqueous-phase systems (e.g., fuel cells that use hydrazine and sodium borohydride in alkaline solutions as the fuel) [46]. Subsequent trials were conducted using the anhydrous cement sample at a relative humidity of 100% and 70 °C.

### 3.2. Ionic Conducting Characteristics of Portland Cement

In the ionic conduction model described in the previous section, not only OH^−^ ions, but also metal (Ca^2+^, Na^+^, or K^+^) ions are possible charge carriers. A series of trials were made to determine the essential charge carrier in the cement sample. The cement sample was galvanostatically polarized with a current density of 0.4 mA cm^−2^ in humidified air for 60 h. Assuming that Ca^2+^ or Na^+^ (or K^+^) ions are the charge carrier in the sample, the mole number of metal migrated from the anode to the cathode over 60 h was calculated to be 2.2 × 10^−4^ moles for Ca or 4.4 × 10^−4^ moles for Na according to Faraday’s law. Thus, 8.97 mg (25% of sample weight) of Ca or 10.3 mg (29% of sample weight) of Na would be expected to be consumed at the anode and produced at the cathode. SEM images of the sample surfaces at the anode and cathode sides after polarization are presented in Figure 5, along with the corresponding EDX spectra. No significant difference in morphology was observed between the two surfaces. In addition, the magnitude of the EDX signal for Ca or Na at the cathode side was comparable to that at the anode side. Thus, the major charge carrier in the sample was determined to be OH^−^ ions rather than Ca^2+^ or Na^+^ ions, because both oxygen and water vapor, which are the source of OH^−^ ions, are continuously supplied to the cathode. This postulation is supported by the appearance of XRD peaks for calcium carbonate (CaCO_3_) at the anode side, which was in contrast to the appearance of XRD peaks for Ca(OH)_2_ at the cathode side (Figure 6). In the case of a hydroxide ion conductor, carbon corrosion occurs at the anode by polarization at high electrode potentials [47,48], so that Ca(OH)_2_ reacts with CO_2_ produced according to Reaction (4) to form CaCO_3_.
Anode: C (electrode) + O_2_ (gas) → CO_2_ (gas).(4)

Two types of galvanic cells, hydrogen and oxygen concentration cells (hereafter denoted as H_2_ and O_2_ cells, respectively), were fabricated using the cement sample as the electrolyte, and measurements were conducted at 70 °C. The electromotive force (EMF) values were recorded while varying the partial pressure of hydrogen (*P*_H2_) at the cathode in the H_2_ cell with *P*_H2_ at the anode in the H_2_ cell kept constant and by varying the partial pressure of oxygen (*P*_O2_) at the anode in the O_2_ cell with *P*_O2_ at the cathode in the O_2_ cell kept constant (see the Method section for *P*_H2O_ at all electrodes). The theoretical EMF values (*Es*) of the H_2_ and O_2_ cells were calculated according to the following equations [33,34]:

H_2_ cell:

*P*_H2 High_, Pt/C | Cement sample | Pt/C, *P*_H2 Low_,
Anode: H_2_ + 2OH^−^ → 2H_2_O + 2e^−^,(5)Cathode: 2H_2_O + 2e^−^ → 2OH^−^ + H_2_,(6)*E* = *RT*/2*F* ln(*P*_H2 Low_/*P*_H2 High_).(7)

O_2_ cell:

*P*_O2 Low_, Pt/C | Cement sample | Pt/C, *P*_O2 High_,
Anode: 4OH^−^ → 2H_2_O + O_2_ + 4e^−^,(8)Cathode: 2H_2_O + O_2_ + 4e^−^ → 4OH^−^,(9)*E* = *RT*/4*F* ln(*P*_O2 High_/*P*_O2 Low_),(10)
where *R*, *T*, and *F* denote the molar gas constant, the absolute temperature, and Faraday’s constant, respectively. However, it should be noted that since Equations (7) and (10) are also derived by assuming that metal ions are the charge carrier, the EMF is not necessarily generated by a certain specific ionic conduction. Figure 7 shows that the observed EMF values were almost in agreement with the theoretical values for both the H_2_ and O_2_ cells, which indicates that the total ion transport number of this sample was approximately 1 in both the oxidative and reductive atmospheres. Therefore, this sample did not exhibit p- or n-type semiconductivity under such conditions, which is very convenient for use in electrochemical applications [49].

### 3.3. Degradation of Ionic Conduction in Cement by CO_2_ and Subsequent Amelioration

Cement pastes have a high risk of having their chemical and mechanical characteristics altered in the presence of both CO_2_ and water [50,51]. CO_2_ is dissolved in water to form carbonic acid (H_2_CO_3_), which penetrates into the cement, dissolving Ca(OH)_2_ into Ca^2+^ and OH^−^ ions. Ca^2+^ ions and OH^−^ ions easily react with hydrogen carbonate and H^+^ to form CaCO_3_ and water, respectively. Similar carbonation can also occur for the present cement sample, because a small amount of Ca(OH)_2_ was produced through Reactions (1) and (2) under humidified conditions. The impedance spectra were measured in a humidified Ar flow containing 0.5% CO_2_. Figure 8a shows a significant decrease in the bulk and interfacial conductivities with the CO_2_ exposure time, the effect of which was larger for the interface than for the bulk. An XRD pattern was obtained after exposure to CO_2_ and subsequent drying. As shown in Figure 8b, the formation of CaCO_3_ was confirmed, and the peak intensities for C_3_S and C_2_S were decreased, which is due to the dissolution of parts of these components in the carbonic acid. These results demonstrate a decrease in the amount of OH^−^ ions, which reduce both the bulk and interfacial conductivities. In addition, CaCO_3_ precipitates on the surface of C_3_S and C_2_S particles or on the pore wall of C-S-H products, which further reduces the interfacial conductivity. 

The degradation of ionic conduction by CO_2_ is commonly encountered in anion exchange membranes, although the mechanism for carbonation is slightly different among the various materials [52,53]. However, there have been no reports on the recovery of ionic conductivity in a carbonated membrane. Thus, an attempt was made to ameliorate the ionic conductivity of the carbonated sample by immersion in 18 mol% acetic acid solution at room temperature (the concentration of 18% is not an optimized value). A large amount of fine bubbles were formed from the sample surface in the solution. After bubble formation was completed, the sample was thoroughly washed with deionized water and then dried in the drying container. As a result of the acid treatment, the bulk conductivity returned to the initial value or greater, while the interfacial conductivity was improved to 17% of the initial value (Figure 8a). The XRD peaks of CaCO_3_ disappeared along with those of C_3_S, C_2_S, and C_3_A, while those of C_4_AF remained after the acid treatment (Figure 8b). The Rietveld analysis (Figure 8c) revealed that the octahedral and tetrahedral layers were alternately connected in the C_4_AF structure to form a three-dimensional framework with the Ca^2+^ ions in interstitial voids (space group *Ibm2*, a = 5.5283(5) Å, b = 14.573(1) Å, and c = 5.3272(5) Å), as shown in Figure 8d. Ectors et al. recently proposed new reaction steps for the C_4_AF hydration process, which is accompanied by the production of Ca(OH)_2_ [6]. Thus, it is possible that similar to C_3_S and C_2_S, C_4_AF provides conductive OH^−^ ions to the sample through Reactions (1) and (2), which returns the bulk conductivity to the initial value. On the other hand, since Na_2_O and K_2_O are most likely washed away with deionized water in the acid treatment, the contribution of these metal ions to the bulk conductivity is negligible under the present conditions. A possible reason for the interfacial conductivity being lower than the initial value is that insulating materials are formed at the interface between the C_4_AF particles by the acid treatment; the XRD pattern of the sample after acid treatment and drying had an elevated background in the range 20–40° (Figure 8b), which indicates a non-negligible amount of amorphous materials [54]. Therefore, this technique should be further optimized by precise control of the amount of acid or the treatment time.

## 4. Conclusions

Ionic conduction in anhydrous cement particles and hydrated cement pastes was investigated by preparing a 300 μm thick membrane sample, followed by EIS measurements in an atmosphere saturated with water vapor (relative humidity = 100%) at various temperatures (50–100 °C). The ionic conductivity consisted of bulk and interfacial components, distinguished by the ion transport that corresponds to the frequency and dependent on the structural factors or not. Both the conductivity components were enhanced by an increase in the *P*_H2O_ and temperature, the effects of which were attributed to the reaction of CaO in the C_3_S and water vapor at high *P*_H2O_ and to the acceleration of this reaction at high temperature, respectively. An additional effect was obtained by the formation of ion channels arising from the aggregated water molecules through the cement. Crystalline Ca(OH)_2_ was not necessarily required for ionic conduction in the cement, thus allowing for the use of anhydrous cement particles as an ionic conductor. OH^−^ ions functioned as the charge carrier in the cement, and no change in structure or composition by long-term polarization of the sample was observed. Other charge carriers, such as Ca^2+^, Na^+^, and K^+^ ions, electrons, and electron holes, were excluded, at least under the present conditions. Similar to conventional hydroxide ionic conductors, the cement suffered from a degradation of ionic conduction in the presence of CO_2_ and water vapor through the dissolution of CO_2_ in water, and the subsequent formation of CaCO_3_. The lowered bulk and interfacial conductivities were recovered to some extent by a decarbonation reaction in acetic acid solution at room temperature.

## Figures and Tables

**Figure 1 materials-12-03701-f001:**
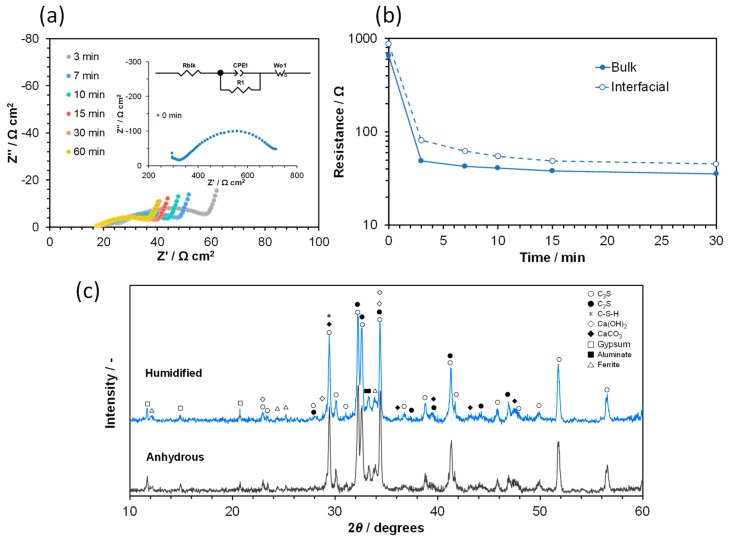
Characterization of the cement sample humidified in an atmosphere saturated with water vapor at 50 °C. (**a**) Impedance spectra recorded at various times after switching *P*_H2O_ from 0.03 to 0.12 atm. The inset shows the equivalent circuit model used to fit the impedance data. (**b**) Bulk and interfacial resistances computed from the equivalent circuit as a function of the time after switching *P*_H2O_. (**c**) XRD patterns before and after humidification and drying.

**Figure 2 materials-12-03701-f002:**
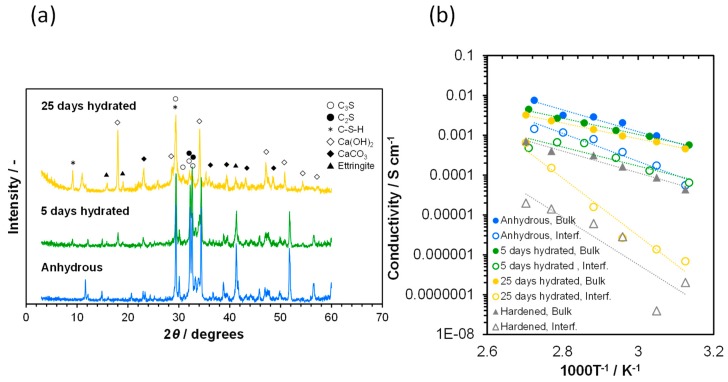
Characterization of the cement samples with different degrees of hydration. (**a**) XRD patterns. (**b**) Temperature dependence of the bulk and interfacial conductivities.

**Figure 3 materials-12-03701-f003:**
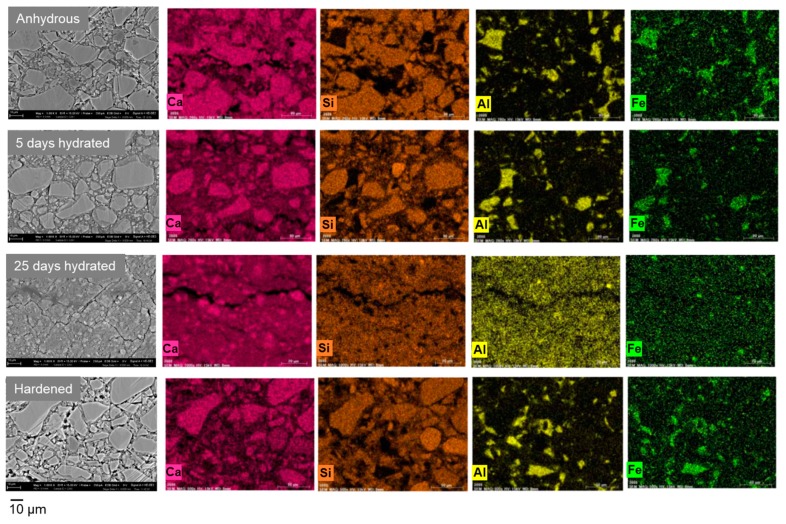
SEM images and energy dispersive X-ray (EDX) elemental maps for Ca, Si, Al, and Fe on the cross section of anhydrous, 5 days and 25 days hydrated, and hardened hydrated cement samples.

**Figure 4 materials-12-03701-f004:**
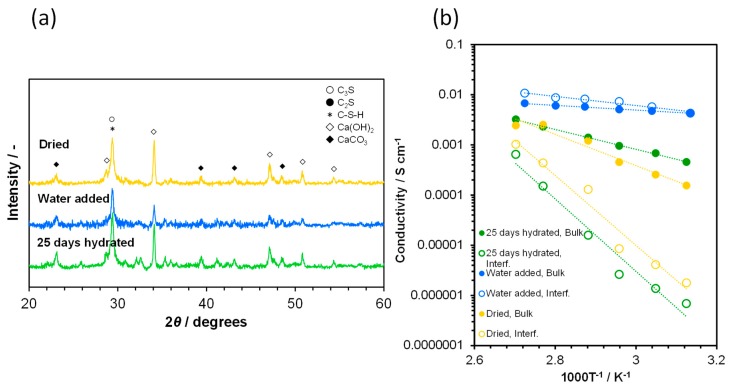
Characterization of the highly hydrated sample before and after addition of 15% water, and after drying. (**a**) XRD patterns. (**b**) Temperature dependence of the bulk and interfacial conductivities.

**Figure 5 materials-12-03701-f005:**
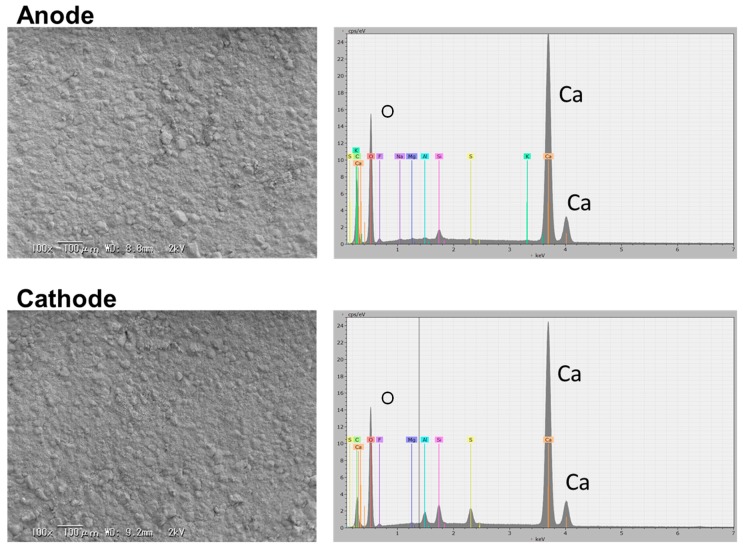
SEM images and EDX element analysis of the sample surfaces at the anode and cathode sides after cell polarization for 60 h. Cell polarization was conducted in humidified air at 70 °C.

**Figure 6 materials-12-03701-f006:**
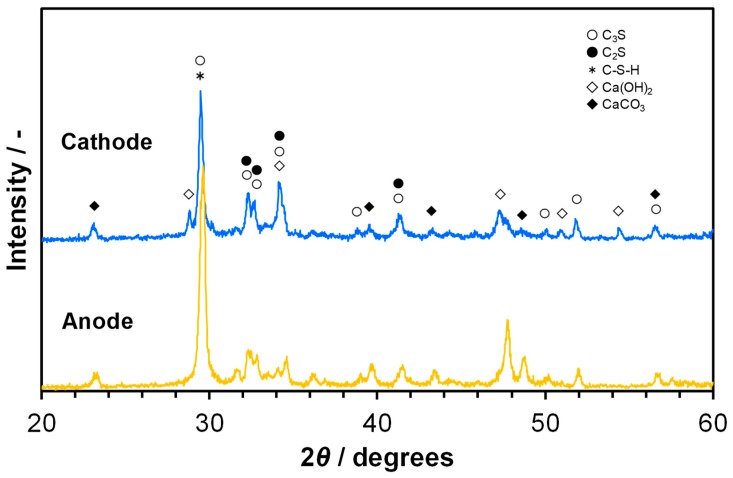
XRD profiles after polarization with a current density of 0.4 mA cm^−2^ in humidified air at 70 °C for 60 h, followed by drying in a CO_2_-free container.

**Figure 7 materials-12-03701-f007:**
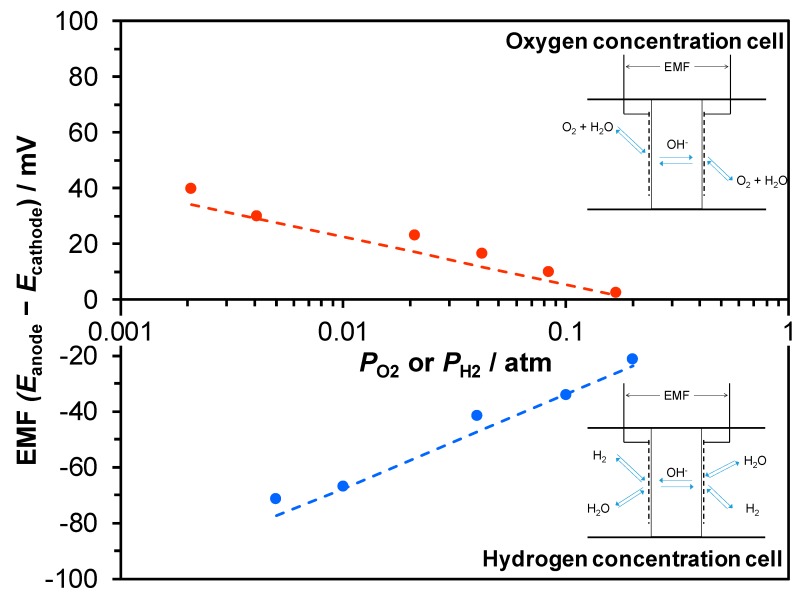
Electromotive forces (EMFs) generated from hydrogen and oxygen concentration cells at 70 °C as a function of *P*_H2_ and *P*_O2_, respectively. The dotted lines show the theoretical values for each gas concentration cell.

**Figure 8 materials-12-03701-f008:**
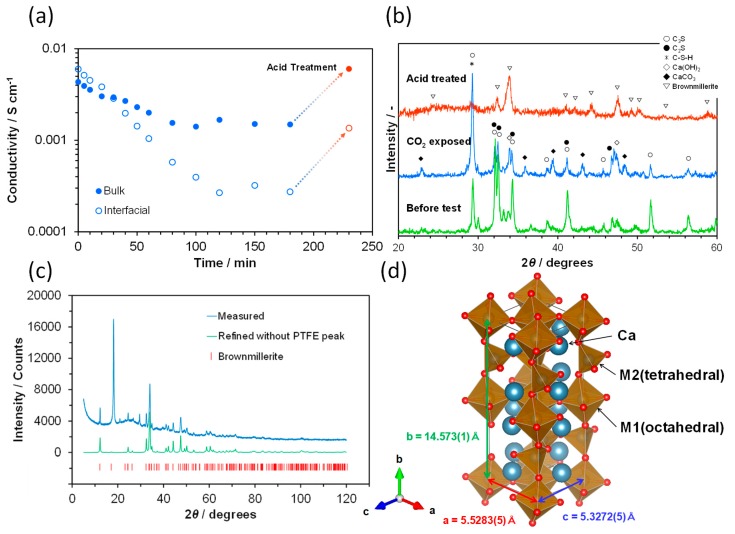
(**a**) Bulk and interfacial conductivities recorded at various times after exposure to a mixture of CO_2_ and water vapor at 70 °C, and after subsequent acid treatment. (**b**) XRD patterns for the corresponding cement samples in Figure 6a. (**c**) Rietveld refined XRD pattern of the sample after the acid treatment. (**d**) Crystal structure of C_4_AF (brownmillerite) drawn using the Rietveld refinement results.

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
