# Peer review of "Humidity Driven Transition from Insulator to Ionic Conductor in Portland Cement"

_materials, 2019, doi:10.3390/ma12223701_

Round 1

Reviewer 1 Report

This is an interesting paper, dealing with a topic appropriate for the Journal.

It is well written and well organized. However, some modifications/additions are necessary before its acceptance for publication.

The authors should clarify the type of the various materials and samples investigated. Furthermore, appropriate terminology concerning the cementitious mixes should be used. The terms clinker and cement must not be used interchangeably, the terms anhydrous clinker (or cement) particles, hydrated cement pastes or dried hydrated cement pastes should be used in place of clinker and humidified.

Also, the characteristics of the sample indicated as “clinker” should be clarified. Based on the description reported at page 4, line 125, it appears to be a not anhydrous sample.

See the attached file for specific comments.

Author Response

Responses to Reviewer 1

This is an interesting paper, dealing with a topic appropriate for the Journal. It is well written and well organized. However, some modifications/additions are necessary before its acceptance for publication.

Response: We are delighted to read the referee’s positive evaluation for our work. Based on the comments from the reviewer, we have made the suggested corrections and changes.

The authors should clarify the type of the various materials and samples investigated. Furthermore, appropriate terminology concerning the cementitious mixes should be used. The terms clinker and cement must not be used interchangeably, the terms anhydrous clinker (or cement) particles, hydrated cement pastes or dried hydrated cement pastes should be used in place of clinker and humidified.

Response: According to the reviewer’s comment, all the terms of the used and tested samples have been corrected. Please refer to the following point-by-point responses to specific comments.

Page 1, line 8: “Portland cements with different degrees of hydration” was corrected to “anhydrous cement particles and hydrated cement pastes with aging periods of 5 and 25 days”

Page 1, lines 29-31: “The cement paste formed from clinker and water eventually maintains a pH in the range of 11–13 due to the large amount of Ca(OH)2 produced, until it is depleted” was corrected to “Similar to sodium hydroxide (NaOH) and potassium hydroxide (KOH) present in the hydrated cement paste, the produced Ca(OH)2 results in a pH level (11–13) in the cement paste, until these metal hydroxides are”

Page 1, line 38: “The electrical resistance of Portland cement” was corrected to “The electrical resistance of hardened Portland cement pastes”.

Pages 1-2, lines 44-45: “the measurement of Portland cement” was corrected to “the measurement of cement particles and hydrated cement pastes”.

Page 2, lines 69-70: “A mortar cube” was corrected to “A hardened hydrated cement paste cube”.

Page 2, line86: “The pH of the cement sample” was corrected to “The pH of the cement sample immediately after humidification”.

Figure 1d: According to the comment, this figure was omitted from Fig. 1.

Page 4, lines 128-129: “obtained after aging at PH2O = 0.03 atm and 50 °C” was corrected to “obtained after previously aging at PH2O = 0.03 atm and 50 °C”. “Exactly speaking, at this stage, the sample has already been hydrated with the 0.03 atm water vapor.”

Page 4, lines 133-134: “that of the virgin clinker sample” was corrected to “that of the anhydrous cement sample”.

Page 4, lines 136-137: The virgin cement particles contain small amounts of CaSO4·2H2O and alkaline metal oxides (Na2O, and K2O) [?]. CaSO4·2H2O is the origin of gypsum. This description about the composition was added to the revised manuscript.

Page 4, line 156: “the solubility of Ca(OH)2 is as low as 0.13 g per 100 g water at 50 °C.” was ambiguous, as pointed out by the reviewer. This description was omitted in the revised manuscript.

Since then, the terms of the used and tested samples have been corrected as well. The corrections were highlighted in the revised manuscript.

Reviewer 2 Report

The paper presents ionic conduction in Portland cement with different degrees of hydration.

The topic of the study is interesting and relevant from an academic and industrial point of view. However:

1/ The work is written in an incomprehensible and lengthy manner, and therefore it is difficult to understand and connect all the parts of the work

2/ No explanations of the changes of the ionic conduction were given by the chemical and physical properties of the mortar. Therefore, an essential goal of academic and scientific understanding is lacking.

3/ The meaning of the hydration degree was not explained and calculated, as it should be.

4/ The measurement of the conduction or the resistivity of the cement mortar is well known. Not as claimed on Page 1. In the building industry, various properties are obtained by measuring the resistivity of the cement mortar, as chloride penetration, compressive strengths and setting times. However, understanding the changes in the electrochemical properties are not well known, and was expected to be explored in the current paper deeply.

5/ Why water to cement ratio of 1/3 was chosen? The testing procedures and the selected materials should be explained.

6/ It is difficult to understand the figures given. The figures were not well explained and understood.

As was said, the article is written in a very complex and incomprehensible way, and therefore it was complicated to read and to understand it.

Author Response

Responses to Reviewer 2

The paper presents ionic conduction in Portland cement with different degrees of hydration. The topic of the study is interesting and relevant from an academic and industrial point of view. However:

Response: We are delighted to read the referee’s very positive evaluation for our work. Based on the comments from the reviewer, we have modified the manuscript and Figures.

1/ The work is written in an incomprehensible and lengthy manner, and therefore it is difficult to understand and connect all the parts of the work

Response: Based on this comment, we have rewritten many contents of the manuscript as clearly and shortly as possible. For example, the contribution of NaOH and KOH present in the cement materials to ionic conduction, which was somewhat ambiguous in the original manuscript, was discussed in more detail (Pages 1, 4, and 7). Also, the names of samples treated in different manners were unified according to the comments from Reviewer 1. These corrections were highlighted in the revised manuscript.

2/ No explanations of the changes of the ionic conduction were given by the chemical and physical properties of the mortar. Therefore, an essential goal of academic and scientific understanding is lacking.

Response: The term “mortar” described in the original manuscript was incorrect. The corresponding sample is a hardened hydrated cement paste. However, if the reviewer made this comment for all samples rather than mortar, please note that we made characterization that is thought to be necessary for explanation of ionic conduction in the field of solid state ionics.

3/ The meaning of the hydration degree was not explained and calculated, as it should be.

Response: The hydration degree was adjusted by control of the mixing time, which was reflected by the intensity change of the peaks assigned to C3S, C2S, C3A, C4AF, Ca(OH)2, and CSH. Thus, since we thought that this change is easily understood and traced, the three degrees of hydration were classified as anhydrous, 5 days and 25 days hydrated states without quantification.

4/ The measurement of the conduction or the resistivity of the cement mortar is well known. Not as claimed on Page 1. In the building industry, various properties are obtained by measuring the resistivity of the cement mortar, as chloride penetration, compressive strengths and setting times. However, understanding the changes in the electrochemical properties are not well known, and was expected to be explored in the current paper deeply.

Response: This information has been added to Introduction in the revised manuscript.

5/ Why water to cement ratio of 1/3 was chosen? The testing procedures and the selected materials should be explained.

Response: This ratio was selected to prepare cement pastes with a good balance between the strength and porosity, as reported elsewhere. This description has been inserted on page 2 in the revised manuscript.

6/ It is difficult to understand the figures given. The figures were not well explained and understood.

Response: The legends of Figs. 1, 2, and 4 were rewritten to understand the data more easily. We believe that figures became easily understood by modifying the text as well as figures.

As was said, the article is written in a very complex and incomprehensible way, and therefore it was complicated to read and to understand it.

Response: This article relates to different fields of cement and concrete materials and solid state ionics. The preparation, operation, and characterization of the cement samples may be unusual in the former field, but normal when considering the application of this membrane to a solid electrolyte for fuel cells and electrolyzers. Further work is needed to bridge the gap between the fields.

Round 2

Reviewer 2 Report

The author presented a revised and improved paper. However, the comments provided in the first version of the article have not yet been fully revised.

Author Response

Responses to Reviewer 2

The author presented a revised and improved paper. However, the comments provided in the first version of the article have not yet been fully revised.

Response: Based on this comment, we have revised the manuscript as follows.

3/ The meaning of the hydration degree was not explained and calculated, as it should be.

Response: We measured the weights of the chemically bound water for two hydrated cement samples by using TG analysis and calculated their degrees of hydration from the data according to the method reported in Ref. [32]. The procedures and the results have been described on pages 2 and 4, respectively, in the revised manuscript.

4/ The measurement of the conduction or the resistivity of the cement mortar is well known. Not as claimed on Page 1. In the building industry, various properties are obtained by measuring the resistivity of the cement mortar, as chloride penetration, compressive strengths and setting times. However, understanding the changes in the electrochemical properties are not well known, and was expected to be explored in the current paper deeply. 

Response: This information has been added to Introduction in the revised manuscript, along with the corresponding references [15-18].

5/ Why water to cement ratio of 1/3 was chosen? The testing procedures and the selected materials should be explained.

Response: This ratio was selected to prepare cement pastes with a good balance between the strength and porosity, as reported elsewhere [31]. This description has been inserted on page 2 in the revised manuscript.